# FleXOR: Trainable Fractional Quantization

## Abstract

Parameter quantization is a popular model compression technique due to its regular form and high compression ratio. In particular, quantization based on binary codes is gaining attention because each quantized bit can be directly utilized for computations without dequantization using look-up tables. Previous attempts, however, only allow for integer numbers of quantization bits, which ends up restricting the search space for compression ratio and accuracy. Moreover, quantization bits are usually obtained by minimizing quantization loss in a local manner that does not directly correspond to minimizing the loss function. In this paper, we propose an encryption algorithm/architecture to compress quantized weights in order to achieve fractional numbers of bits per weight and new compression configurations further optimize accuracy/compression trade-offs. Decryption is implemented using XOR gates added into the neural network model and described as $\tanh(x)$, which enable gradient calculations superior to the straight-through gradient method. We perform experiments using MNIST, CIFAR-10, and ImageNet to show that inserting XOR gates learns quantization/encrypted bit decisions through training and obtains high accuracy even for fractional sub 1-bit weights.

## 1 Introduction

Deep Neural Networks (DNNs) demand larger number of parameters and more computations to support various task descriptions all while adhering to ever-increasing model accuracy requirements. Because of abundant redundancy in DNN models (Han et al., 2016; Frankle & Carbin, 2019; Courbariaux et al., 2015) and limited scaling technology of integrated circuits due to physical challenges, numerous model compression techniques are being aggressively studied to expedite inference of DNNs across a range of device scales. As a practical model compression scheme, parameter quantization is a popular choice because compression ratio is high and regular formats after compression enable full memory bandwidth utilization that is crucial in highly parallel computing systems.

Quantization schemes based on binary codes are gaining increasing attention since quantized weights follow specific constraints to allow simpler computations during inference. Specifically, using binary codes, a weight vector is represented as $\sum_{i=1}^{q}(\alpha_i \sum_{j=1}^{v} b_{i,j})$, where $q$ is the number of quantization bits, $v$ is the vector size, $\alpha$ is a scaling factor ($\alpha \in \mathbb{R}$), and $b_{i,j}$ is a binary code ($b_{i,j} \in \{-1, +1\}$). Then, a dot product with activations is conducted as $\sum_{i=1}^{q}(\alpha_i \sum_{j=1}^{v} a_j b_{i,j})$, where $a_j$ is a full-precision activation. Then, the number of multiplications is reduced from $v$ to $q$ (expensive floating-point multipliers are less required for inference). Moreover, even though we do not discuss a new activation quantization method in this paper, if activations are also quantized by using binary codes, then most computations are replaced with bit-wise operations (using XNOR logic and population counts) (Xu et al., 2018; Rastegari et al., 2016). Consequently, even though representation space is constrained compared with quantization methods based on look-up tables, various inference accelerators can be designed to exploit advantages of binary codes (Rastegari et al., 2016; Xu et al., 2018). Since a successful 1-bit weight quantization method has been demonstrated in BinaryConnect (Courbariaux et al., 2015), advances in compression-aware training algorithms in the form of binary codes (e.g., binary weight networks (Rastegari et al., 2016) and LQ-Nets (Zhang et al., 2018)) produce 1-3 bits for quantization while accuracy drop is modest or negligible. Fundamental investigations on DNN training mechanisms using fewer quantization bits have also been actively reported to improve quantization quality (Li et al., 2017).

Figure 1: FleXOR components added to the quantized DNNs to compress quantized weights through encryption. Encrypted weight bits are decrypted by XOR gates to produce quantized weight bits.

Previously, binary-coding-based quantization has only permitted integer numbers of quantization bits, limiting the compression/accuracy trade-off search space, especially in the range of very low quantization bits. In this paper, we propose a flexible encryption algorithm/architecture (called "FleXOR") to enable fractional sub 1-bit numbers to represent each weight while quantizated bits are trained by gradient descent. Even though vector quntization is also a well-known scheme with high compression ratio (Stock et al., 2019), we assume the form of binary codes. Note that the number of quantization bits can be different for each layer (e.g., Wang et al. (2019)) to allow fractional quantization bits on average. FleXOR implies fractional quantization bits for each layer that can be quantized with different bits.

The main contributions of this paper are as follows:

**XOR-based encryption of quantized bits enhances compression ratio:** We encrypt quantized weights to further compress quantized DNNs applying basic principles from cryptography and digital communication.

**XOR-aware training algorithm learns encrypted weights:** A binary model of XOR gates is used for forward propagation and model inference. We suggest calculating gradients of XOR gates using $\tanh$ functions instead of using a straight-through estimator (STE) (Bengio et al., 2013) for efficient compression-aware training.

**High model accuracy with sub 1-bit quantization:** To the best of our knowledge, *our work is the first to explore model accuracy under 1 bit/weight when weights are quantized based on binary codes.* Such an exploration plays a significant role for mobile devices where it may be possible to sacrifice modest amounts of accuracy drop in exchange for improved energy efficiency.

## 2 ENCRYPTING QUANTIZED BITS USING XOR GATES

The main purpose of FleXOR is to compress quantized bits into encrypted bits that can be reconstructed by XOR gates as shown in Figure 1. Suppose that $N_{out}$ bits are to be compressed into $N_{in}$ bits ($N_{out} > N_{in}$). The role of an XOR-gate network is to produce various $N_{out}$-bit combinations using $N_{in}$ bits. In other words, in order to maximize the chance of generating a desirable set of quantized bits, the encryption scheme is designed to seek a particular property where all possible $2^{N_{in}}$ outcomes through decryption are evenly distributed in $2^{N_{out}}$ space.

A linear Boolean function, $f(\boldsymbol{x})$, maps $f : \{0,1\}^{N_{in}} \rightarrow \{0,1\}$ and has the form of $a_1 x_1 \oplus a_2 x_2 \oplus \cdots \oplus a_{N_{in}} x_{N_{in}}$ where $a_j \in \{0,1\}$ ($1 \le j \le N_{in}$) and $\oplus$ indicates bitwise modulo-2 addition. In Figure 1, six binary outputs are generated through six Boolean functions using four binary inputs. Let $f_1(\boldsymbol{x})$ and $f_2(\boldsymbol{x})$ be two such linear Boolean functions using $\boldsymbol{x} = (x_1, x_2, ..., x_{N_{in}}) \in \{0,1\}^{N_{in}}$. The Hamming distance between $f_1(\boldsymbol{x})$ and $f_2(\boldsymbol{x})$ is the number of inputs on which $f_1(\boldsymbol{x})$ and $f_2(\boldsymbol{x})$ differ, and defined as

$$d_H(f_1, f_2) := w_H(f_1 \oplus f_2) = \#\{\boldsymbol{x} \in \{0,1\}^{N_{in}} | f_1(\boldsymbol{x}) \neq f_2(\boldsymbol{x})\}, \qquad (1)$$

where $w_H(f) = \#\{\boldsymbol{x} \in \{0,1\}^{N_{in}} | f(\boldsymbol{x}) = 1\}$ is the Hamming weight of a function and $\#\{\}$ corresponds to the size of a set (Kahn et al., 1988). The Hamming distance is a well-known method to express non-linearity between two Boolean functions (Kahn et al., 1988) and increased Hamming

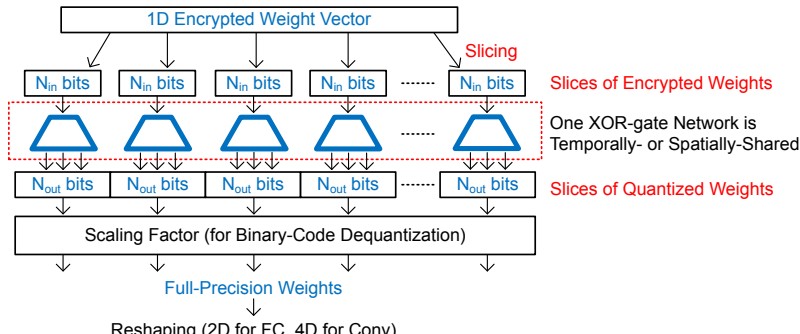

Figure 2: Encrypted weight bits are sliced and reconstructed by a XOR-gate network which can be shared (in time or space). Then quantized bits after XOR gates are finally reshaped.

distance between a pair of two Boolean functions results in a variety of outputs produced by XOR gates. Increasing Hamming distance is a required feature for cryptography to derive complicated encryption structure such that inverting encrypted data becomes difficult. For digital communication, the Hamming distance between encoded signals is closely related to the amount of error correction possible.

In Figure 1, $y_1$ is represented as $x_1 \oplus x_3 \oplus x_4$, or equivalently a vector $[1\,0\,1\,1]$ denoting which inputs are selected. Concatenating such vectors, a XOR-gate network in Figure 1 can be described as a binary matrix $\boldsymbol{M}^{\oplus} \in \{0,1\}^{N_{out} \times N_{in}}$ (e.g., the second row of $\boldsymbol{M}^{\oplus}$ is $[1\,1\,0\,0]$ and the third row is $[1\,1\,1\,0]$). Then, decryption through XOR gates is simply represented as $\boldsymbol{y} = \boldsymbol{M}^{\oplus}\boldsymbol{x}$ where $\boldsymbol{x}$ and $\boldsymbol{y}$ are the binary inputs and binary outputs of XOR gates, and addition is 'XOR' and multiplication is 'AND' (see Appendix for more details and examples).

Encrypted weight bits are stored in 1-dimensional vector format and sliced into blocks of $N_{in}$-bit size as shown in Figure 2. Then, decryption of each slice is performed by a XOR-gate network that is shared by all slices (temporally- or spatially-shared). Depending on the quantization scheme and characteristics of layers, quantized bits may need to be scaled by a scaling factor and/or reshaped.

## 3    FleXOR Training Algorithm for Quantization Bits Decision

Once the structure of XOR gates has been pre-determined and fixed to increase the Hamming distance of XOR outputs, we find quantized and encrypted bits by adding XOR gates into the model. In other words, we want an optimizer that understands the XOR-gate network structure so as to compute encrypted bits and scaling factors via gradient descent. Note that for inference, we store binary encrypted weights (converted from real number encrypted weights) in memory and generate binary quantized weights through Boolean XOR operations. Activation quantization is not discussed in this paper.

Similar to the STE method introduced in (Courbariaux et al., 2015), Boolean functions need to be described in a differentiable manner to obtain gradients in backward propagation. For two real number inputs $x_1$ and $x_2$ ($x_1, x_2 \in \mathbb{R}$ to be used as encrypted weights), the Boolean version of a XOR gate for forward propagation is described as (note that 0 is replaced with $-1$)

$$\mathcal{F}^{\oplus}(x_1, x_2) = (-1)\,\mathrm{sign}(x_1)\,\mathrm{sign}(x_2). \tag{2}$$

For inference, we store $\mathrm{sign}(x_1)$ and $\mathrm{sign}(x_2)$ instead of $x_1$ and $x_2$. On the other hand, a differentiable XOR gate for backward propagation is presented as

$$f^{\oplus}(x_1, x_2) = (-1)\tanh(x_1 \cdot S_{\tanh})\tanh(x_2 \cdot S_{\tanh}), \tag{3}$$

where $S_{\tanh}$ is a scaling factor for FleXOR. Note that $\tanh$ functions are widely used to approximate Heaviside step functions (i.e., $y(x) = 1$ if $x > 0$ or 0, otherwise) in digital signal processing and $S_{\tanh}$ can control the steepness (as explained in Appendix). In the case of consecutive XOR operations, the order of inputs to be fed into XOR gates should not affect the computation of partial

gradients for XOR inputs. Therefore, as a simple extension of Eq. (3), a differentiable XOR gate network with $n$ inputs can be described as

$$f^{\oplus}(x_1, x_2, \ldots, x_n) = (-1)^{n-1} \tanh(x_1 \cdot S_{\tanh}) \tanh(x_2 \cdot S_{\tanh}) \ldots \tanh(x_n \cdot S_{\tanh}). \quad (4)$$

Then, a partial derivative of $f^{\oplus}$ with respect to $x_i$ (an encrypted weight) is given as

$$\frac{\partial f^{\oplus}(x_1, x_2, \ldots, x_n)}{\partial x_i} = S_{\tanh}(-1)^{n-1}(1 - \tanh^2(x_i \cdot S_{\tanh}))\frac{\prod_{j=1}^n \tanh(x_j \cdot S_{\tanh})}{\tanh(x_i \cdot S_{\tanh})} \quad (5)$$

Note that increasing the Hamming distance is associated with more $\tanh$ multiplications for each XOR-gate network output. Then, from Eq. (5), we may suffer from the vanishing gradient problem since $|\tanh(x)| \leq 1$. To resolve this problem, we also consider a simplified alternative partial derivative expressed as

$$\frac{\partial f^{\oplus}(x_1, x_2, \ldots, x_n)}{\partial x_i} \approx S_{\tanh}(-1)^{n-1}(1 - \tanh^2(x_i \cdot S_{\tanh}))\prod_{j \neq i} \text{sign}(x_j). \quad (6)$$

Eq. (6) shows that when we compute a partial derivative, all XOR inputs other than $x_i$ are assumed to be binary, i.e., the magnitude of a partial derivative is then only determined by $x_i$. We use Eq. (6) in this paper to calculate custom gradients of encrypted weights due to fast training computations and convergence, and use Eq. (2) for forward propagation.

---

**Algorithm 1:** Pseudo codes of Conv Layer with FleXOR[1]

---

$\boldsymbol{w}^e \in R^{\lceil (k \cdot k \cdot C_{in} \cdot C_{out})/N_{out} \rceil \cdot N_{in}}$         ▷ Encrypted weights

$\boldsymbol{M}^{\oplus} \in \{0, 1\}^{N_{out} \times N_{in}}$         ▷ XOR gates (shared)

$\boldsymbol{\alpha} \in \mathbb{R}^{C_{out}}$         ▷ Scaling factors for each output channel

**Function** `FleXOR_Conv` *(input, stride, padding)* **:**

    **for** $i \leftarrow 0$ **to** $\lceil (k \cdot k \cdot C_{in} \cdot C_{out})/N_{out} \rceil - 1$ **do**

        **for** $j \leftarrow 1$ **to** $N_{out}$ **do**

$$w^q_{i \cdot N_{out}+j} \leftarrow (-1) \cdot \left( \prod_{\substack{l=1 \\ M^{\oplus}_{j,l}=1}}^{N_{in}} \texttt{Sign}^c \left( w^e_{i \cdot N_{in}+l} \right) \cdot (-1) \right) \qquad \text{▷ Eq. (2)}$$

    $\mathbf{W}^q \leftarrow \texttt{Reshape}(\boldsymbol{w}^q, [k, k, C_{in}, C_{out}])$

    **return** `Conv`*(input,* $\mathbf{W}^q$*,* $\boldsymbol{\alpha}$*, stride, padding)*      ▷ Convolution operation for binary codes

**Forward Function** `Sign`$^c$ *(x)* **:**

    **return** $\text{sign}(x)$

**Gradient Function** `Sign`$^c$ *(x, $\nabla$)* **:**

    **return** $\nabla \cdot (1 - \tanh^2(x \cdot S_{\tanh})) \cdot S_{\tanh}$      ▷ Eq. (6)

---

By training the whole network including FleXOR components with using custom gradient computation methods described above, encrypted and quantized weights are obtained in a holistic manner. FleXOR operations for convolutional layers are described in Algorithm 1, where encrypted weights (inputs of a XOR-gate network) and quantized weights (outputs of a XOR-gate network) are $\boldsymbol{w}^e$ and $\mathbf{W}^q$. We first verify basic training principles of FleXOR using LeNet-5 on the MNIST dataset. LeNet-5 consists of two convolutional layers and two fully-connected layers (specifically, 32C5-MP2-64C5-MP2-512FC-10SoftMax), and each layer is accompanied by an XOR-gate network with $N_{in}$ binary inputs and $N_{out}$ binary outputs. The quantization scheme follows 1-bit binary code with full-precision scaling factors that are shared across weights for the same output channel number (for conv layers) or output neurons (for FC layers). Encrypted weights are randomly initialized with $\mathcal{N}(\mu=0, \sigma^2=0.001^2)$. All scaling factors for quantization are initialized to be 0.1 (note that if batch normalization layers are immediately followed, then scaling factors for quantization are redundant).

---

[1]Kernel size is $k \times k$, the number of input channel and output channel are $C_{in}$ and $C_{out}$, respectively

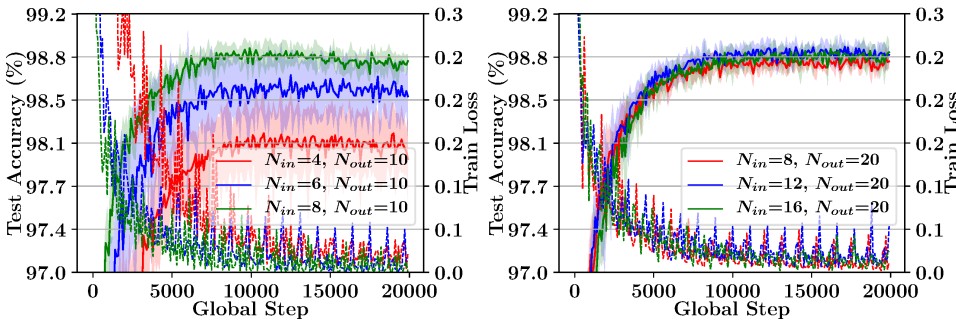

Figure 3: Test accuracy and training loss (average of 6 runs) with LeNet-5 on MNIST when $M^\oplus$ is randomly filled with $\{0, 1\}$. $N_{out}$ is 10 or 20 to generate, effectively, 0.4, 0.6, or 0.8 bit/weight quantization.

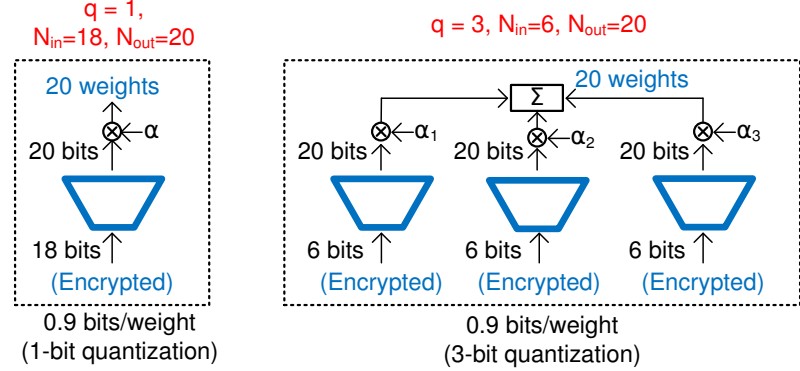

Figure 4: Using the same weight storage footprint, FleXOR enables various internal quantization schemes. (Left): 1-bit internal quantization. (Right): 3-bit internal quantization with 3 different $M^\oplus$ configurations.

Using the Adam optimizer with an initial learning rate of $10^{-4}$ and batch size of 50 without dropout, Figure 3 shows training loss and test accuracy when $S_{\tanh}=100$, elements of $M^\oplus$ are randomly filled with 1 or 0, and for two values of $N_{out}$ – 10 and 20. Using the 1-bit internal quantization method and $(N_{in}, N_{out})$ encryption scheme, one weight can be represented by $(N_{in}/N_{out})$ bits. Hence, Figure 3 represents training results for 0.4, 0.6, and 0.8 bits per weight. Note that as for a randomly filled $M^\oplus$, increasing $N_{out}$ (and $N_{in}$ correspondingly for the same compression ratio) increases the Hamming distance for a pair of any two rows of $M^\oplus$ and, hence, offers the chance to produce more diversified outputs. Indeed, as shown in Figure 3, the results for $N_{out}=20$ present improved test accuracy and less variation compared with $N_{out}=10$. See Appendix for the distribution of encrypted weights at different training steps.

## 4 PRACTICAL FLEXOR TRAINING TECHNIQUES

In this section, we present practical training techniques for FleXOR using ResNet-32 (He et al., 2016) on the CIFAR-10 dataset (Krizhevsky et al., 2009). As shown in Figure 4, since FleXOR decouples the number of bits to represent weights from the quantization scheme used, various trade-offs between memory footprint and model accuracy are possible. We show compression results for ResNet-32 using fractional numbers as effective quantization bits, such as 0.4 and 1.2, that have not been available previously.

All layers, except the first and the last layers, are followed by FleXOR components sharing the same $M^\oplus$ structure (thus, storage foorprint of $M^\oplus$ is ignorable). SGD optimizer is used with momentum of 0.9 and a weight decay factor of $10^{-5}$. Initial learning rate is 0.1, which is decayed by 0.5 at the 150th and 175th epoch. As learning rate decays, $S_{\tanh}$ is empirically multiplied by 2 to cancel

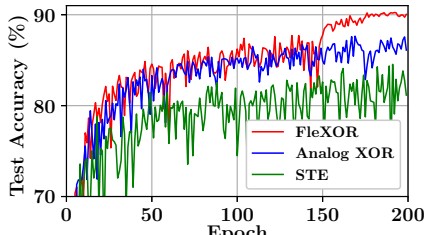

| | XOR Training Method | | |
|---|---|---|---|
| | STE | Analog XOR | FleXOR |
| Forward | sign | tanh | sign |
| Backward | Identity | $\partial(\tanh)$ | $\partial(\tanh)$ |
| XOR | Binary | $\mathbb{R}$ | Binary |
| Output | $-1$ or $+1$ | $(-1, +1)$ | $-1$ or $+1$ |

Figure 5: Test accuracy comparison on ResNet-32 (for CIFAR-10) using various XOR training methods. $N_{out} = 10$, $N_{in} = 8$, $q = 1$ (thus, 0.8bit/weight), and $S_{\tanh} = 10$.

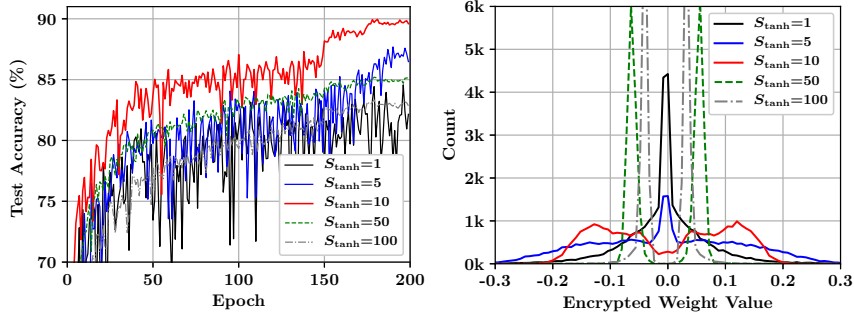

Figure 6: Test accuracy and distribution of encrypted weights (at the end of training) of ResNet-32 on CIFAR-10 using various $S_{\tanh}$ and the same $N_{out}$, $N_{in}$, and $q$ as Figure 5.

out the effects of weight decay on encrypted weights. Batch size is 128 and scaling factors of $\alpha$ are randomly initialized between 0.01 and 0.19. $q$ is the number of bits to represent binary codes for quantization. We provide some useful training insights below with relevant experimental results.

1) **Use small $N_{tap}$ (such as 2):** FleXOR should be able to select the best out of $2^{N_{in}}$ possible outputs that are randomly selected from larger $2^{N_{out}}$ search space. Encryption performance of XOR gates is determined by randomness of $2^{N_{in}}$ output candidates, and is enhanced by increasing Hamming distance that is basically achieved by large $N_{out}$. Now, let $N_{tap}$ be the number of 1's in a row of $M^{\oplus}$. Another method to enhance encryption performance is to increase $N_{tap}$ so as to increase the number of shuffles (through more XOR operations) using encrypted bits to generate quantized bits such that correlation between quantized bits is reduced. Large $N_{tap}$, however induces vanishing gradient problems in Eq. (5) or increased approximation error in Eq. (6). Hence, in practice, FleXOR training with small $N_{tap}$ converges well with high test accuracy. Studying a training algorithm to understand a complex XOR-gate network with large $N_{tap}$ would be an interesting research topic that is beyond the scope of this work. Subsequently, we show experimental results using $N_{tap} = 2$ in the remainder of this paper.

2) **Use 'tanh' rather than STE:** Since forward propagation for a XOR gate only needs a sign function, the STE method is also applicable to XOR-gate gradient calculations. Another alternative method to model an XOR gate is to use Eq. (3) for both forward and backward propagation as if the XOR is modeled in an analog manner (then, real number XOR outputs are quantized through STE). We compare three different XOR modeling schemes in Figure 5 with test accuracy measured when encrypted weights and XOR gates are converted to be binary for inference. FleXOR training method shows the best result because a) sign function for forward propagation enables estimating the impact of binary XOR computations on the loss function and b) $\partial(\tanh)$ for backward propagation approximates the Heaviside step function better compared to STE. Note that limited gradients from the tanh function eliminate the need for weight clipping, which is often required for quantization-aware training schemes (Courbariaux et al., 2015; Xu et al., 2018).

3) **Optimize $S_{\tanh}$:** $S_{\tanh}$ controls the smoothness of the tanh function for near-zero inputs. Large $S_{\tanh}$ employs large gradient for small inputs and, hence, results in well-clustered encrypted weight

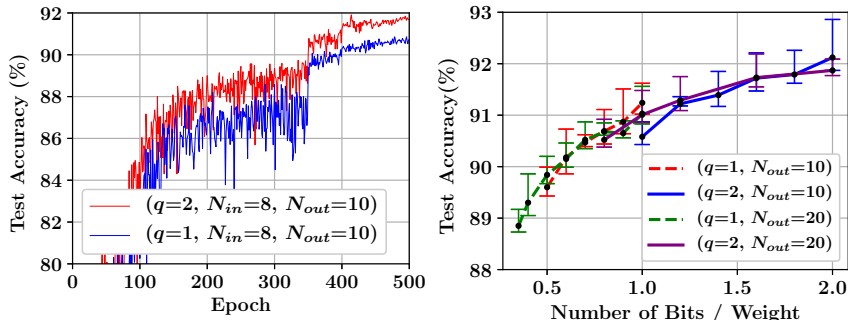

Figure 7: Test accuracy of ResNet-32 on CIFAR10 using learning rate warmup and various $q$, $N_{in}$, and $N_{out}$. The results on the right side are obtained by 5 runs.

Table 1: Weight compression comparison of ResNet-20 and ResNet-32 on CIFAR-10. For FleXOR, we use warmup scheme, $S_{\tanh}$=10, and $N_{out}$=20. All of quantization schemes in the table follow the form of binary codes with $q$=1, and hence, the amount of computations becomes the same. FleXOR, however, saves on/off-chip memory requirements further becasue $N_{in}/N_{out}$ is less than 1.

|  | ResNet-20 | | | ResNet-32 | | |
|---|---|---|---|---|---|---|
|  | FP | Compressed | Diff. | FP | Compressed | Diff. |
| BWN (1 bit) | 92.68%[2] | 87.44% | -5.24% | 93.40%[2] | 89.49% | -4.51% |
| BinaryRelax (1 bit) | 92.68%[2] | 87.82% | -4.86% | 93.40%[2] | 90.65% | -2.80% |
| LQ-Net (1 bit) | 92.10%[3] | 90.10% | -1.90% | - | - | - |
| FleXOR (1.0 bit) |  | 90.44% | -1.47% |  | 91.36% | -0.97% |
| FleXOR (0.8 bit) | 91.87%[4] | 89.71% | -2.10% | 92.33%[4] | 90.78% | -1.55% |
| FleXOR (0.6 bit) |  | 89.04% | -2.83% |  | 90.30% | -2.03% |
| FleXOR (0.4 bit) |  | 87.85% | -4.02% |  | 89.61% | -2.72% |

values as shown in Figure 6. Too large of a $S_{\tanh}$, however, hinders encrypted weights from being finely-tuned through training. For FleXOR, $S_{\tanh}$ is a hyper-parameter to be optimized.

4) **Learning rate and $S_{\tanh}$ warmup:** Learning rate starts from 0 and linearly increases to reach the initial learning rate at a certain epoch as a warmup. Learning rate warmup is a heuristic scheme, but being widely accepted to improve generalization capability mainly by avoiding large learning rate in the initial phase (He et al., 2018; Gotmare et al., 2019). Similarly, $S_{\tanh}$ starts from 5 to linearly increases to 10 using the same warmup schedule of the learning rate.

5) **Try various $q$, $N_{in}$, and $N_{out}$:** Using a warmup scheme for 100 epochs and learning rate decay by 50% at the 350[th], 400[th], and 450[th] epoch, Figure 7 presents test accuracy of ResNet-32 with various $q$, $N_{in}$, and $N_{out}$. For $q>1$, different $M^{\oplus}$ configurations are constructed and then shared across all layers. Note that even for the 0.4bit/weight configuration (using $q$=1, $N_{in}$=8, and $N_{out}$=20), high accuracy close to 89% is achieved. 0.8bit/weight can be achieved by two different configurations (as shown on the right side of Figure 7) using ($q$=1, $N_{in}$=8, $N_{out}$=10) or ($q$=2, $N_{in}$=8, $N_{out}$=20). Interestingly, those two configurations show almost the same test accuracy, which implies that FleXOR is able to provide a linear relationship between the amount of encrypted weights and model accuracy (regardless of internal configurations). In general, lowering $q$ reduces the amount of computations with quantized weights and a high $N_{out}$ value is necessary for very low number of bits per weight since $N_{in}$ also needs to be high enough to increase the Hamming distance.

We compare quantization results of ResNet-20 and ResNet-32 on CIFAR-10 using different compression schemes in Table 1 (with full-precision activation). BWN (Rastegari et al., 2016), BinaryRelax (Yin et al., 2018), and LQ-Net (Zhang et al., 2018) propose different training algorithms

---

[2]Reported by BinaryRelax paper (Yin et al., 2018)

[3]Reported by LQ-Net paper (Zhang et al., 2018)

[4]Based in *https://github.com/akamaster/pytorch_resnet_cifar10*, where we replaced the ReLU functions with leaky ReLU functions.

Table 2: Weight compression comparison of ResNet-18 on ImageNet using various compression schemes. All of quantization schemes in the table follow the form of binary codes with $q=1$, and hence, the amount of computations becomes the same. FleXOR, however, saves on/off-chip memory requirements further becasue $N_{in}/N_{out}$ is less than 1.

| Methods | Bits/Weight | Top-1 | Top-5 | Storage Saving |
|---|---|---|---|---|
| Full Precision (He et al., 2016) | 32 | 69.6% | 89.2% | $1\times$ |
| BWN (Rastegari et al., 2016) | 1 | 60.8% | 83.0% | $\sim 32\times$ |
| ABC-Net (Lin et al., 2017) | 1 | 62.8% | 84.4% | $\sim 32\times$ |
| BinaryRelax (Yin et al., 2018) | 1 | 63.2% | 85.1% | $\sim 32\times$ |
| FleXOR ($N_{out} = 20$) | 0.8 | 63.8% | 84.8% | $\sim 40\times$ |
| | 0.6 | 62.0% | 83.7% | $\sim 53\times$ |
| | 0.5 | 61.3% | 83.1% | $\sim 64\times$ |

for the same quantization scheme (i.e., binary codes). The main idea of these methods is to minimize quantization error and to obtain gradients from full-precision weights while the loss function is aware of quantization. Because all of quantization schemes in Table 1 uses $q=1$ and binary codes, the amount of computations using quantized weights is the same. FleXOR, however, allows reduced memory footprint and bandwidth which are critical to improve energy efficiency of inference engine designs (Han et al., 2016; Ahn et al., 2019). Note that even though achieving the best accuracy for 1.0bit/weight is not the main purpose of FleXOR (e.g., XOR gate may be redundant for $N_{in}=N_{out}$), FleXOR shows the minimum accuracy drop for ResNet-20 and ResNet-32 as shown in Table 1, probably because a) FleXOR enables model optimization integrated with quantization and encryption units instead of considering the local quantization error and b) FleXOR calculates gradients for encrypted weights by employing 'tanh' (instead of STE) that approximates a Heaviside step function.

## 5 EXPERIMENTAL RESULTS ON IMAGENET

In order to show that FleXOR principles can be extended to larger models, we choose ResNet-18 on ImageNet (Russakovsky et al., 2015). We use SGD optimizer with momentum of 0.9 and initial learning rate of 0.1. Batch size is 128, weight decay factor is $10^{-5}$, and $S_{\text{tanh}}$ is 10. Learning rate is reduced by half at the 70[th], 100[th], and 130[th]. For warmup, during initial ten epochs, $S_{\text{tanh}}$ and learning rate increase linearly from 5 and 0.0, respectively, to initial values.

Table 2 shows the comparison on model accuracy of ResNet-18 when weights are compressed by quantization (and additional encryption by FleXOR) while activations maintain full precision. Training the entire model for ImageNet including FleXOR components is successfully performed. In Table 2, BinaryRelax and BWN do not modify the underlying model architecture, while ABC-Net introduces a new block structure of the convolution for quantized network designs. FleXOR achieves the best top-1 accuracy even with only 0.8bit/weight and demonstrates improvement of model accuracy as the number of bits per weight increases. Refer to Appendix for the accuracy graph with $q=1$ and more results with $q=2$.

## 6 CONCLUSION

This paper proposes an encryption algorithm/architecture, FleXOR, as a framework to further compress quantized weights. Encryption is designed to produce more outputs than inputs by increasing the Hamming distance of output functions when output functions are linear functions of inputs. Output functions are implemented as a combination of XOR gates which are included in the model to find encrypted and quantized weights through gradient descent while using the tanh function for backward propagation. FleXOR is able to provide fractional numbers of bits for weights and, thus, much wider trade-offs between weight storage and model accuracy. Experimental results show that ResNet on CIFAR-10 and ImageNet can be represented by sub 1-bit/weight compression with high accuracy.

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

# A APPENDIX

## A.1 EXAMPLE OF A XOR-GATE NETWORK STRUCTURE REPRESENTATION

In Figure 1, outputs of a XOR-gate network are given as

$$y_1 = x_1 \oplus x_3 \oplus x_4$$
$$y_2 = x_1 \oplus x_2$$
$$y_3 = x_1 \oplus x_2 \oplus x_3$$
$$y_4 = x_3 \oplus x_4$$
$$y_5 = x_2 \oplus x_4$$
$$y_6 = x_2 \oplus x_3 \oplus x_4.$$

Equivalently, the same structure as above can be represented in a matrix as

$$
\boldsymbol{M}^{\oplus} =
\begin{bmatrix}
1 & 0 & 1 & 1 \\
1 & 1 & 0 & 0 \\
1 & 1 & 1 & 0 \\
0 & 0 & 1 & 1 \\
0 & 1 & 0 & 1 \\
0 & 1 & 1 & 1
\end{bmatrix}.
\tag{7}
$$

Note that elements of $\boldsymbol{M}^{\oplus}$ are matched with coefficients of $y_i (1 \leq i \leq 6)$. For two vectors $\boldsymbol{y} = \{y_1, y_2, y_3, y_4, y_5, y_6\}$ and $\boldsymbol{x} = \{x_1, x_2, x_3, x_4\}$, the following equation holds:

$$\boldsymbol{y} = \boldsymbol{M}^{\oplus} \cdot \boldsymbol{x}, \tag{8}$$

where element-wise addition and multiplication are performed by 'XOR' and 'AND' function, respectively. In Eq. (7), $N_{tap}$ (i.e., the number of '1's in a row) is 2 or 3.

## A.2 SUPPLEMENTARY DATA FOR BASIC FLEXOR TRAINING PRINCIPLES

A Boolean XOR gate can be modeled as $\mathcal{F}^{\oplus}(x_1, x_2) = (-1) \operatorname{sign}(x_1) \operatorname{sign}(x_2)$ if 0 is replaced with $-1$ as shown in Table 3.

| $\operatorname{sign}(x_1)$ | $\operatorname{sign}(x_2)$ | $\mathcal{F}^{\oplus}(x_1, x_2)$ |
|:---:|:---:|:---:|
| $-1$ | $-1$ | $-1$ |
| $-1$ | $+1$ | $+1$ |
| $+1$ | $-1$ | $+1$ |
| $+1$ | $+1$ | $-1$ |

Table 3: An XOR gate modeling using $\mathcal{F}^{\oplus}(x_1, x_2)$.

In Eq. (7), forward propagation for $y_3$ is expressed as

$$y_3 = \mathcal{F}^{\oplus}(x_1, x_2, x_3) = (-1)^2 \operatorname{sign}(x_1) \operatorname{sign}(x_2) \operatorname{sign}(x_3). \tag{9}$$

while partial derivative of $y_3$ with respect to $x_1$ is given as (not derived from Eq. (9))

$$\frac{\partial y_3}{\partial x_1} = S_{\text{tanh}}(-1)^2 (1 - \tanh^2(x_1 \cdot S_{\text{tanh}})) \tanh(x_2 \cdot S_{\text{tanh}}) \tanh(x_3 \cdot S_{\text{tanh}}), \tag{10}$$

or as

$$\frac{\partial y_3}{\partial x_1} \approx S_{\text{tanh}}(-1)^2 (1 - \tanh^2(x_1 \cdot S_{\text{tanh}})) \operatorname{sign}(x_2) \operatorname{sign}(x_3). \tag{11}$$

We choose Eq. (11), instead of Eq. (10), as explained in Section 3.

As shown in Figure 8, large $S_{\text{tanh}}$ yields sharp transitions for near-zero inputs. Such a sharp approximation of the Heaviside step function produces large gradient values for small inputs and encourages encrypted weights to be separated into negative or positive values. Too large $S_{\text{tanh}}$, however, has the same isseus of a too large learning rate.

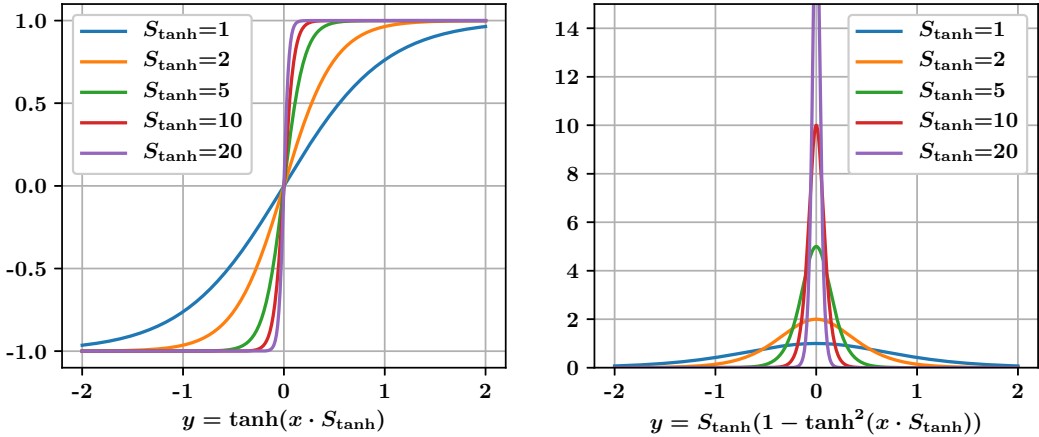

Figure 8: The left graph shows hyperbolic tangent ($y = \tanh(x \cdot S_{\text{tanh}})$) graphs with various scaling factors ($S_{\text{tanh}}$), . The right graph shows their derivatives. These graphs support the arguments of '**Optimize $S_{\text{tanh}}$**' in Section 4.

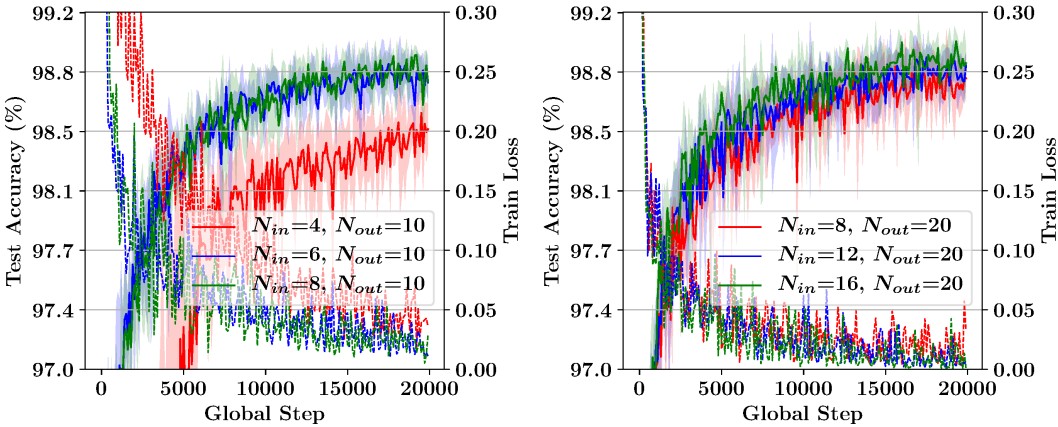

Figure 9: Test accuracy and training loss of LeNet-5 on MNIST when number of '1's in each row of $\boldsymbol{M}^{\oplus}$ is fixed to be 2 ($N_{tap} = 2$). $N_{out}$ is 10 or 20 to generate, effectively, 0.4, 0.6, or 0.8 bit/weight quantization. With low $N_{tap}$ of $\boldsymbol{M}^{\oplus}$, MNIST training presents less variations on training loss and test accuracy that in Figure 3.

Figure 9 presents training loss and test accuracy when $N_{tap} = 2$ and $N_{out}$ is 10 or 20. Compared with Figure 3, $N_{tap} = 2$ presents improved accuracy for the cases of high compression configurations (e.g., $N_{in} = 4$ and $N_{out} = 10$). We use $N_{tap} = 2$ for CIFAR-10 and ImageNet, since low $N_{tap}$ avoids gradient vanishing problems or high approximation errors in Eq.(5) or Eq.(6).

Figure 10 plots the distribution of encrypted weights at different training steps when each row of $\boldsymbol{M}^{\oplus}$ is randomly assigned with $\{0, 1\}$ (i.e., $N_{tap}$ is $N_{in}/2$ on average) or assigned with only two 1's ($N_{tap}$=2). Due to gradient calculations based on $\tanh$ and high $S_{\text{tanh}}$, encrypted weights tend to be clustered on the the left or right (near-zero encrypted weights become less as $N_{tap}$ increases) even without weight clipping.

### A.3 SUPPLEMENTARY EXPERIMENTAL RESULTS OF CIFAR-10 AND IMAGENET

In this subsection, we additionally provide various graphs and accuracy tables for ResNet models on CIFAR10 and ImageNet. We also present experimental results from wider hyper-parameters searches including $q = 2$ with two separate $\boldsymbol{M}^{\oplus}$ configurations (with the same $N_{in}$ and $N_{out}$ for two $\boldsymbol{M}^{\oplus}$ matrices).

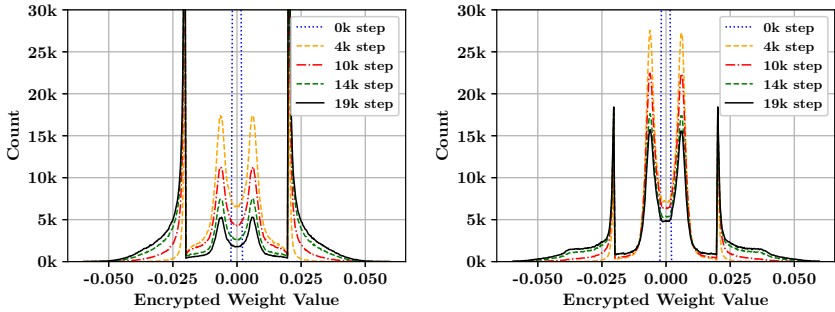

Figure 10: Distribution of encrypted weight values for FC1 layer of LeNet-5 at different training steps using $S_{\tanh} = 100$ and $N_{out} = 10$. (Left): $\boldsymbol{M}^{\oplus}$ is randomly filled ($N_{tap} \approx N_{in}/2$). (Right): $N_{tap} = 2$ for every row of $\boldsymbol{M}^{\oplus}$.

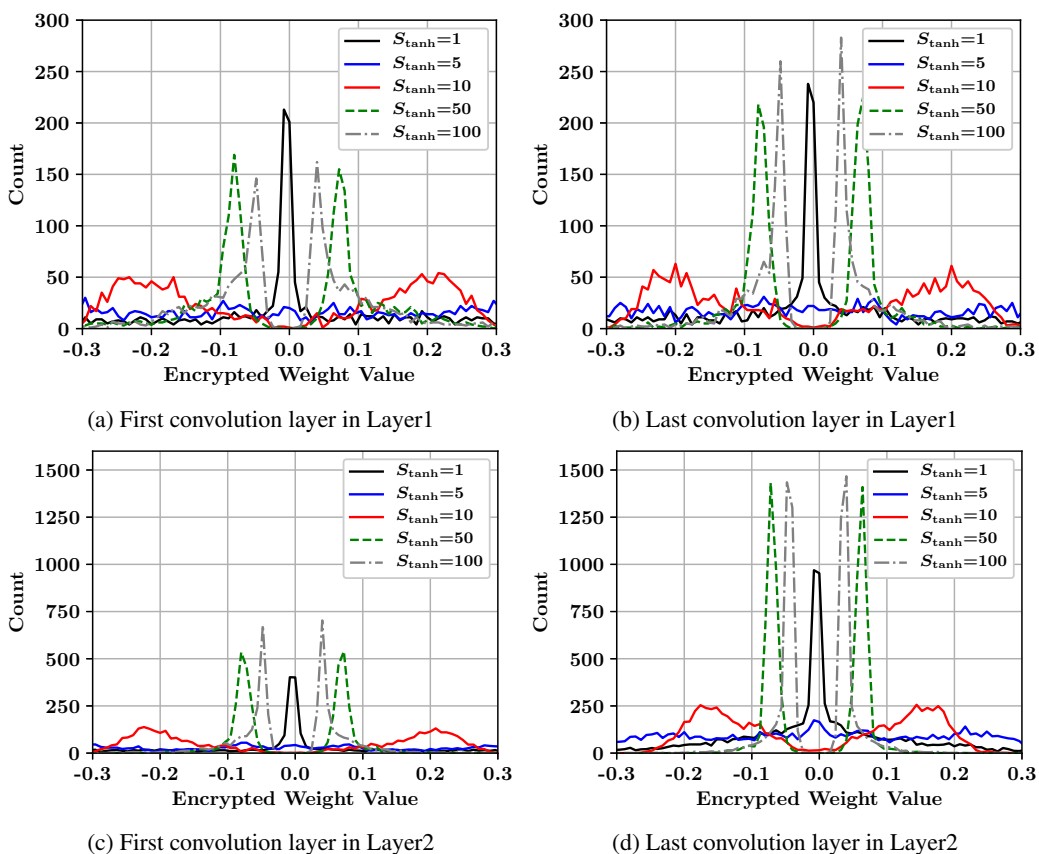

Figure 11: Distributions of encrypted weights (at the end of training) in various layers of ResNet-32 on CIFAR-10 using various $S_{\tanh}$ and the same $N_{out}$, $N_{in}$, and $q$ as Figure 5. The ResNet-32 network mainly consists of three layers according to the feature map sizes: Layer1, Layer2 and Layer3.

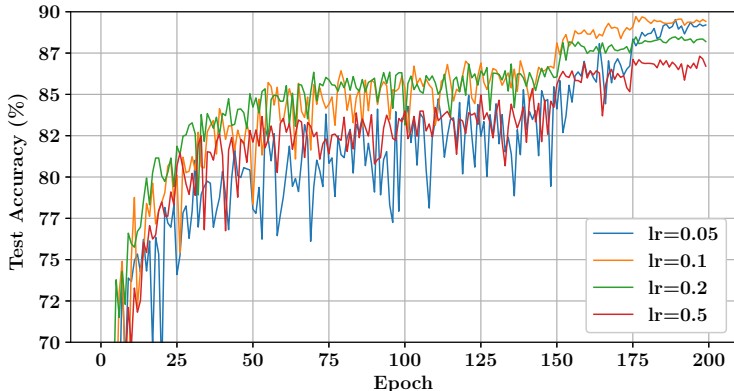

(a) **Initial Learning Rate (0.1)**: Test accuracy of ResNet-32 on CIFAR10 using the learning schedule in Figure 5 and various initial learning rates (0.05, 0.1, 0.2, 0.5).

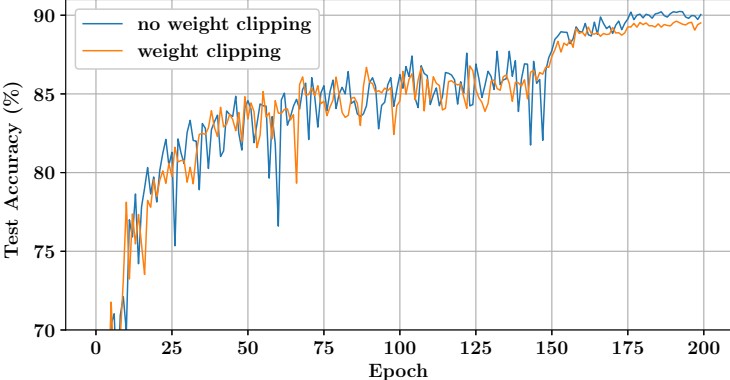

(b) **No Weight Clipping**: Test accuracy of ResNet-32 on CIFAR10 using the learning schedule in Figure 5. As for weight clipping, we restrict the encrypted weights to be ranged as $(-2.0/S_{\text{tanh}}, +2.0/S_{\text{tanh}})$. As can be observed, the red line implies that weight clipping is not effective with FleXOR.

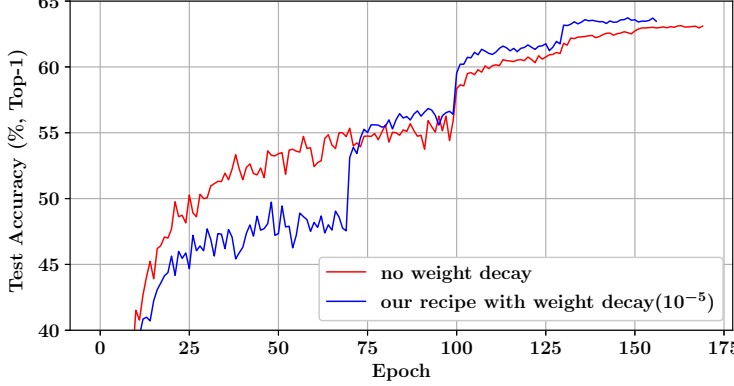

(c) **Weight Decay Factor ($10^{-5}$)**: Two graphs depict test accuracy of ResNet-18 on ImageNet with or without weight decay. The learning rate in the red line (no weight decay) is reduced by half at the $100^{\text{th}}$, $130^{\text{th}}$ and $150^{\text{th}}$ epochs. The learning rate of the blue line (with weight decay) is reduced by half at $70^{\text{th}}$, $100^{\text{th}}$ and $130^{\text{th}}$ epochs. With weight decay (blue graph), despite slow convergence in the early training steps, model accuracy is eventually higher than the red one without weight decay scheme.

Figure 12: Comparison of various hyper-parameter choices for CIFAR-10 or ImageNet.

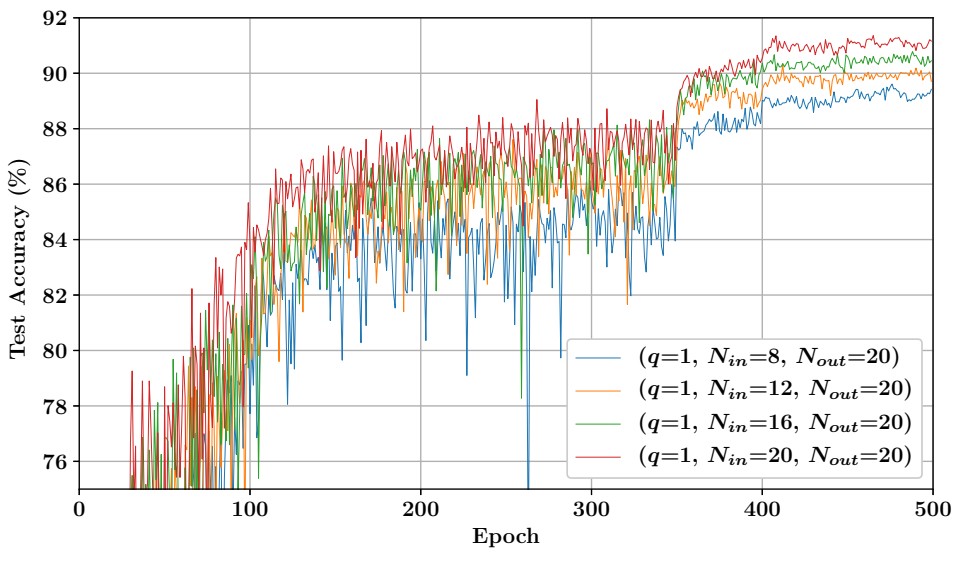

(a) Test accuracy using $q = 1$.

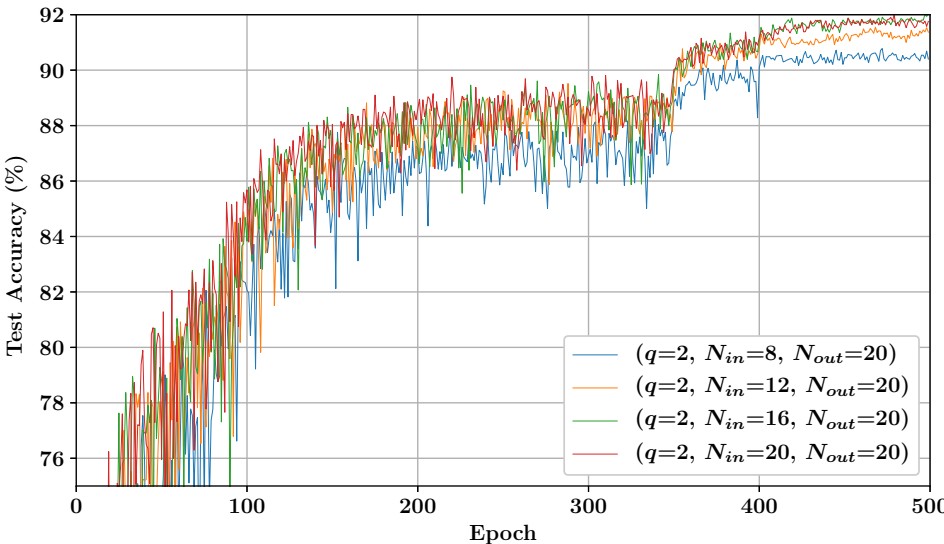

(b) Test accuracy using $q = 2$. Compared to the above plots (Figure 13a), this figure shows that a combination of multiple $M^\oplus$ for a binary code can lead to stable learning curves and higher model accuracy.

Figure 13: Test accuracy of ResNet-32 on CIFAR-10 using learning rate warmup (for 100 epochs) and $N_{out} = 20$

| | Bits/Weight | ResNet-20 | | ResNet-32 | | Comp. Ratio |
|---|---|---|---|---|---|---|
| FP | 32 | 91.87% | - | 92.33% | - | 1.0x |
| $N_{in}$=10, $N_{out}$=10 | 1.0 | 90.21% | -1.66% | 91.40% | -0.93% | 29.95× |
| $N_{in}$=9, $N_{out}$=10 | 0.9 | 90.03% | -1.84% | 91.28% | -1.05% | 31.82× |
| $N_{in}$=8, $N_{out}$=10 | 0.8 | 89.59% | -2.28% | 90.86% | -1.47% | 35.32× |
| $N_{in}$=7, $N_{out}$=10 | 0.7 | 89.24% | -2.63% | 90.54% | -1.79% | 39.68× |
| $N_{in}$=6, $N_{out}$=10 | 0.6 | 89.21% | -2.66% | 90.41% | -1.92% | 45.27× |
| $N_{in}$=5, $N_{out}$=10 | 0.5 | 88.54% | -3.33% | 89.82% | -2.51% | 52.70× |

Table 4: Weight compression comparison of ResNet-20 and ResNet-32 on CIFAR-10 when $N_{out} = 10$. Parameters and recipes not described in the table are the same as in Table 1. We also present compression ratio for fractional quantized ResNet-32 when one scaling factor ($\alpha$) is assigned to each output channel.

| | ResNet-20 | | | ResNet-32 | | |
|---|---|---|---|---|---|---|
| | FP | Quant. | Diff. | FP | Quant. | Diff. |
| TWN (ternary) | 92.68% | 88.65% | -4.03% | 93.40% | 90.94% | -2.46% |
| BinaryRelax (ternary) | 92.68% | 90.07% | -1.91% | 93.40% | 92.04% | -1.36% |
| TTQ (ternary) | 91.77% | 91.13% | -0.64% | 92.33% | 92.37% | +0.04% |
| LQ-Net (2 bit) | 92.10% | 91.80% | -0.30% | - | - | - |
| FleXOR($q = 2$, $N_{out} = 20$) | | | | | | |
| $N_{in}$=20, 2.0 bit/weight | | 91.28% | -0.59% | | 91.99% | -0.34% |
| $N_{in}$=18, 1.8 bit/weight | | 90.96% | -0.91% | | 92.01% | -0.32% |
| $N_{in}$=16, 1.6 bit/weight | 91.87% | 90.81% | -1.06% | 92.33% | 91.55% | -0.78% |
| $N_{in}$=14, 1.4 bit/weight | | 90.22% | -1.65% | | 91.33% | -1.00% |
| $N_{in}$=12, 1.2 bit/weight | | 89.88% | -1.99% | | 90.78% | -1.55% |
| FleXOR($q = 2$, $N_{out} = 10$) | | | | | | |
| $N_{in}$=10, 2.0 bit/weight | | 91.06% | -0.81% | | 92.61% | +0.28% |
| $N_{in}$=9, 1.8 bit/weight | | 91.00% | -0.87% | | 92.09% | -0.24% |
| $N_{in}$=8, 1.6 bit/weight | 91.87% | 90.82% | -1.05% | 92.33% | 91.96% | -0.37% |
| $N_{in}$=7, 1.4 bit/weight | | 90.81% | -1.06% | | 91.63% | -0.70% |
| $N_{in}$=6, 1.2 bit/weight | | 90.28% | -1.59% | | 91.32% | -1.01% |

Table 5: Weight compression comparison of ResNet-20 and ResNet-32 on CIFAR-10 using learning rate warmup (for 100 epochs) and $q = 2$. As mentioned in Figure 4, multiple $M^{\oplus}$ can be combined for multi-bit quantization schemes. Then, the number of scaling factors should be doubled. FleXOR with $q = 2$ and two different $M^{\oplus}$ structures achieve full-precision accuracy when both $N_{in}$ and $N_{out}$ are 10.

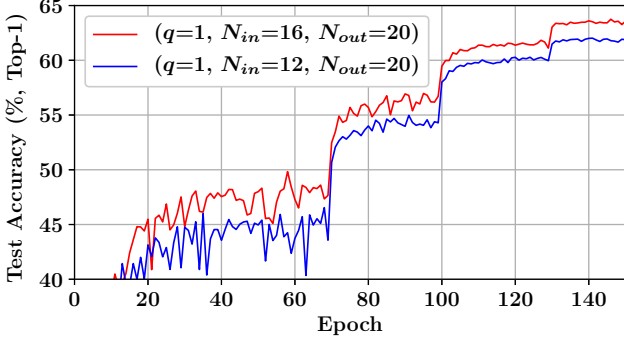

Figure 14: Test accuracy (Top-1) of ResNet-18 on ImageNet using FleXOR.

| Methods | Bits/Weight | Top-1 | Top-5 |
|---|---|---|---|
| Full Precision (He et al., 2016) | 32 | 69.6% | 89.2% |
| TWN (Li & Liu, 2016) | ternary | 61.8% | 84.2% |
| ABC-Net (Lin et al., 2017) | 2 | 63.7% | 85.2% |
| BinaryRelax (Yin et al., 2018) | ternary | 66.5% | 87.3% |
| TTQ(1.5× Wide) (Zhu et al., 2017) | ternary | 66.6% | 87.2% |
| LQ-net (Zhang et al., 2018) | 2 | 68.0% | 88.0% |
| QIL (Jung et al., 2019) | 2 | 68.1% | 88.3% |
| FleXOR ($q = 2$, $N_{out} = 20$) | 1.6 (0.8×2) | 66.2% | 86.7% |
| | 1.2 (0.6×2) | 65.4% | 86.0% |
| | 0.8 (0.4×2) | 63.8% | 85.0% |

Table 6: Weight compression comparison of ResNet-18 on ImageNet when $q = 2$. Since $q$ is 2, we also list the other compression schemes which use 2-bit or ternary quantization scheme for model compression.

