# OpenReview forum: "FleXOR: Trainable Fractional Quantization"
_ICLR.cc/2020/Conference — Reject_

### Official Review · AnonReviewer1 · 2019-10-17
**Official Blind Review #1**

**Rating:** 3

**Review:**

Summary:
The authors propose quantize the weights of a neural network by enabling a fractional number of bits per weight. They use a network of differentiable XOR gates that maps encrypted weights to higher-dimensional decrypted weights to decode the parameters on-the-fly and learn both the encrypted weights and the scaling factors involved in the XOR networks by gradient descent.

Strengths of the paper:
- The method allows for a fractional number of bits per weights and relies of well-known differentiable approximations of the sign function. Indeed, virtually any number of bits/weights can be attained by varying the ratio N_in/N_out.
- The papers displays good results on ImageNet for a ResNet-18.

Weaknesses of the paper:
- Some arguments that are presented could deserve a bit more precision. For instance, quantizing to a fractional number of bits per weights per layer is in itself interesting. However, if we were to quantize different layers of the same network with distinct integer  ratio of bits per weights (say 1 bit per weight for some particular layers and 2 bits per weight for the other layers), the average ratio would also be fractional (see for instance "Hardware-aware Automated Quantization with Mixed Precision", Wang et al., where the authors find the right (integer) number of bits/weights per layer using RL). Similarly, using vector quantization does allow for on-chip low memory: we do not need to re-instantiate the compressed layer but we can compute the forward in the compressed domain (by splitting the activations into similar block sizes and computing dot products).
- More extensive and thorough experiments could improve the impact of the paper. For instance, authors could compress the widely used (and more challenging) ResNet-50 architecture, or try other tasks such as image detection (Mask R-CNN). The table is missing results from: "Hardware Automated Quantization", Wang et al ; "Trained Ternary Quantization", Zhu et al ; "Deep Compression",  Han et al; "Ternary weight networks", Li et al (not an extensive list).
- Similarly, providing some code and numbers for inference time would greatly strengthen the paper and the possible usage of this method by the community. Indeed, I wonder what the overhead of decrypting the weights on-the-fly is (although it only involves XOR operations and products)
- Small typos: for instance, two points at the very end of section 5.

Justification fo rating:
The proposed method is well presented and illustrated. However, I think the paper would need either (1) more thorough experimental results (see comments above, points 2 and 3 of weaknesses) or (2) more justifications for its existence (see comments above, point 1 of weaknesses).

**Experience Assessment:**

I have published one or two papers in this area.

**Review Assessment: Checking Correctness Of Derivations And Theory:**

N/A

**Review Assessment: Checking Correctness Of Experiments:**

I carefully checked the experiments.

**Review Assessment: Thoroughness In Paper Reading:**

I read the paper thoroughly.

---

> ### Author Response · Authors · 2019-11-12
> **Response to AnonReviewer1**
>
> We would like to thank you for the review and comments.
> We revised the manuscript to address your concerns.
> Below we summarized your concerns/questions with our answers.
>
> Q1: Some arguments that are presented could deserve a bit more precision.
> A1: We acknowledge that quantization is a very active research area in model compression and there are numerous quantization techniques with unique and distinct characteristics. We could not introduce and discuss lots of exciting quantization techniques such as vector quantization due to the limited space. We feel that introducing other quantization techniques in details would make the paper distracted since those techniques cannot be compared with compression ratio only (i.e., inference architecture, computation methods, and storage design would be different). Instead, we added more thorough introduction to binary codes in Section 1 to explain unique computational advantages of using binary codes.
> We introduced "Hardware-aware Automated Quantization with Mixed Precision" in Section 1 since fractional quantization on average is available as you pointed out, while FleXOR can also employ different quantization bits for each layer (i.e., we believe HAQ method can be applied on top of FleXOR).
>
> Q2: More extensive and thorough experiments could improve the impact of the paper.
> A2: We agree that including extensive quantization methods and model architectures would greatly improve the impact of the paper. Unfortunately, as we discussed above, our goal in this paper is to improve quantization schemes based on binary codes. Including quantization methods of different assumptions may require much lengthy discussions that make comparisons a lot complicated. For example, "Hardware-aware Automated Quantization" could be additionally applied to binary codes, and FleXOR is not conflicted with such an architectural techniques to improve compression ratio. Deep compression, TTQ, and TWN involve weight pruning that deserves large space for discussions (nonetheless, we compared TWN, TTQ, and BinaryRelax using ternary quantization scheme in Table 5 of Appendix). Deep compression also includes CSR format and Huffman coding which would make comparisons more complicated.
> We chose a few representative quantization methods mainly based on binary codes to facilitate fair and focused comparisons, and correspondingly, ResNet models on CIFAR-10 and ImageNet are selected for our experiments since most previous works (of binary codes) commonly include those models. For example, we could not include HAQ in the paper for experimental results, because HAQ chooses MobileNet and ResNet-50 only as model architectures while comparisons are made with only PACT and Deep Compression methods.
>
> Q3: Providing some code and numbers for inference time would be great.
> A3: Due to the internal policy of our organization, we cannot open our codes publicly at this moment. Hence, we provide a link to anonymous code to the reviewers only until we get an approval for public release. Please refer to our message available to the reviewers only.
> Overhead of weight decryption on-the-fly is extremely small even with CPUs or GPUs, since decryption involves only a binary matrix multiplication over GF(2), which can be easily supported by existing SIMD or vector operations. Since a binary matrix is too small (e.g., 10x8), computational overhead is just ignorable compared with other computations.

---

### Official Review · AnonReviewer2 · 2019-10-22
**Official Blind Review #2**

**Rating:** 6

**Review:**

This paper proposed a fractional quantization method for deep net weights. It adds XOR gates to produce quantized weight bits compared with existing quantization method. It used tanh functions instead of a straight-through estimator for backpropagation during training. With both designs, the proposed method outperformed the existing methods and offered sub bit quantization options.

The use of XOR gates to improve quantization seems novel. The sub bit quantization achieved by this method should be interesting to the industrial. It significantly improved the quantization rate with slightly quality degradation. With 1 bit quantization, it outperformed the state-of-the-art. The results seem thorough and convincing.


**Experience Assessment:**

I do not know much about this area.

**Review Assessment: Checking Correctness Of Derivations And Theory:**

I did not assess the derivations or theory.

**Review Assessment: Checking Correctness Of Experiments:**

I assessed the sensibility of the experiments.

**Review Assessment: Thoroughness In Paper Reading:**

I made a quick assessment of this paper.

---

> ### Author Response · Authors · 2019-11-12
> **Response to AnonReview2**
>
> Response to AnonReviewer2
>
> We would like to thank you for your positive review and comments.
> We would be happy if you have any other questions.

---

### Official Review · AnonReviewer3 · 2019-10-23
**Official Blind Review #3**

**Rating:** 3

**Review:**

The paper proposes a new approach for quantizing neural networks. It looks at binary codes for quantization instead of the usual lookup tables because quantization approaches that rely on lookup tables have the drawback that during inference, the network has to fetch the weights from the lookup table and work with the full precision values, which means that the computation cost is remains the same as the non-quantized network. The paper presents FleXOR gates, a fast and efficient logic gate for mapping bit sequences to binary weights. The benefit of the gate is that the bit sequence can be shorter than the binary weights which means that the code length can be less than 1 bit per weight.

The paper also proposes an algorithm for end-to-end training of the quantized neural networks. It proposes the use of tanh to approximate the non-differentiable Heaviside step function in the backward pass.

Novelty

The idea of using logic gates for dequantization is interesting and (as far as I know) novel. One can imagine, that specialized hardware build on this idea could very efficient for inference (in terms of energy cost).

Writing

The paper is very well written and completed with great visualizations and pseudocode. Kudos to the authors, I really enjoyed reading it. However, I do not think it is justified to go over the 8 page soft limit. I would recommend that the authors perhaps shorten section 3 or remove figure 9 to fit it into 8 pages.

Significance/Impact

The paper is motivated by the high computation cost of working with full precision values. But this paper also works with full precision weights, since it has a full precision scaling factor (alpha) and, as far as I understood, works with full precision values during forward propagation. This means that there likely are no computational savings when compared to lookup tables.

The evaluation section lacks experiments that evaluate the computational savings. The baselines should include quantization methods based on lookup tables, and there should be a comparison of computational costs. The baselines that are presented (BWN etc.) offer a tradeoff between accuracy and computational costs, yet they are only compared in accuracy. I would strongly recommend including the computational cost of each method in the evaluation section.

Overall assessment:
While I enjoyed reading this paper, I am leaning towards rejection due to the shortcomings of the evaluation section.

**Experience Assessment:**

I have read many papers in this area.

**Review Assessment: Checking Correctness Of Derivations And Theory:**

I assessed the sensibility of the derivations and theory.

**Review Assessment: Checking Correctness Of Experiments:**

I carefully checked the experiments.

**Review Assessment: Thoroughness In Paper Reading:**

I read the paper at least twice and used my best judgement in assessing the paper.

---

> ### Author Response · Authors · 2019-11-12
> **Response to AnonReviewer3**
>
> We would like to thank you for the review and comments.
> We revised the manuscript to address your concerns.
> Below we summarized your concerns/questions with our answers.
>
> Q1: It is not justified to go over the 8 page soft limit.
> A1: We removed some redundant information in the paper, moved a few paragraphs and figures to Appendix, and added discussions according to the review comments in the revised manuscript. Now, the paper has full 8 pages that is a soft limit.
>
> Q2: There likely no computational savings when compared to lookup tables.
> A2: As we added in Section 1, if weights are quantized in binary codes, then the number of multiplications is significantly reduced (even though scaling factors have full precision) or most computations can be replaced with bit-wise operations, which have been introduced and discussed as unique advantages of using binary codes in previous works. Since we do not suggest new computation methods using binary codes, computational savings using quantized weights become the same as those of previous binary-codes-based quantization techniques. FleXOR, however, saves on- and off-chip memory requirements significantly if $N_{in}$ is smaller than $N_{out}$, and reducing memory bandwidth/footprint is crucial to designing energy-efficient inference systems. We included this discussion in the evaluation parts.
>
> Q3: The evaluation section lacks experiments that evaluate the computational savings.
> A3: Since binary codes and lookup table would be associated with vastly different inference architecture, computation methods, and storage design, it is difficult to analyze detailed comparisons on FleXOR and lookup-table methods. We chose quantization schemes using binary codes in the experimental results because 1) binary codes are being widely studied and 2) we can focus on the practical issues on binary codes. Since all of quantization techniques in Table 1 and Table 2 follow the form of binary codes with the same q bits, comparisons have been made under the same computational savings (thus, model accuracy is emphasized). FleXOR, however, provides not only higher model accuracy but also additional storage savings due to the proposed encryption algorithm/architecture using XOR logic. We added discussions on the same computational savings and additional storage savings of FleXOR in Section 4 and 5.

---

### Public Comment · ~Jeonghoon_Kim1 · 2019-10-01
**Adding the hardware perspective**

Saving hardware resources in the Edge environment is important.
In this regard, the main idea seems fresh and feasible.

However, it is better to mention the loss of the newly added decryption part and add hardware resource gains through encryption to pictures, tables, or anything because it is adding new processing process to the neural network. (Of course, the hardware loss for the decryption part will not be significant, but it would be nice to mention it.)

This content is also interesting because the encryption and decryption of network weights can be related to security issues in terms of applications.

---

### Author Response · Authors · 2019-11-14
**Revised manuscript is uploaded**

As our responses to reviewers' comments, we uploaded a revised manuscript with the following major changes.

- We removed some redundant information in the paper, moved a few paragraphs and figures to Appendix, and added discussions according to the reviewers' suggestions. Now, the paper has full 8 pages.
- We clarified the computational advantages of binary codes in Section 1 to justify our selection of binary codes as our underlying quantization scheme.
- We added discussions of a few additional relevant papers (including HAQ and vector quantization papers)
- Computational savings and storage savings of FleXOR are discussed in evaluation parts.
- We added a link to FleXOR code available to reviewers (public code release would be available along with the final manuscript submission).

---

### Decision · Program_Chairs · 2019-12-19

**Decision:**

Reject

**Comment:**

This work studies parameter quantization using binary codes and proposes an encryption algorithm/architecture to compress quantized weights and achieve fractional numbers of bits per weight, and to perform decryption using XOR gates. The authors conduct experiments on datasets including ImageNet to evaluate their scheme.
Much of the concern from reviewers relates to baseline comparison and details around that. Specifically, R1 believes that the submission could have a bigger impact if authors could conduct more thorough experiments, e.g. compressing more widely-used and challenging architecture of ResNet-50, or trying tasks such as image detection (Mask R-CNN). The authors' responded to that and mentioned their choice of the current experimental setting is to facilitate comparison with previous works (baselines), which use similar experimental settings. Nevertheless, the baseline methods could have been attempted by the authors on broader tasks, or more widely-used architectures could have been investigated by authors on the baseline methods. As a result, R1 was not convinced. To ensure the paper receives the attention it deserves, I recommend considering a more thorough evaluation of the proposed method against baseline methods.